# Glass factory found: Basinwide (600 km) preservation of sponges on the Phosphoria glass ramp, Permian, USA

Zackery Wistort[1][¤]*, Leif Tapanila[2,3], William Moynihan[2,3], Kathleen Ritterbush[1]

1 Department of Geology and Geophysics, University of Utah, Salt Lake City, Utah, United States of America, 2 Department of Geosciences, Idaho State University, Pocatello, Idaho, United States of America, 3 Idaho Museum of Natural History, Pocatello, Idaho, United States of America

¤ Current address: Florida Atlantic University Harbor Branch, Fort Pierce, Florida, United States of America
* zpwistort@gmail.com

## Abstract

A new analysis of misdiagnosed fossil deposits contextualizes the geologic origin for one of North America's most valuable, but enigmatic, sedimentary units: the Phosphoria Rock Complex (Permian). We describe extensive and repeated deposits of *in situ* marine sponge fossils, previously interpreted as silicified trace fossils, which crop out today as mountain cliffs and widespread landmarks in Nevada, Utah, Idaho, Wyoming, and Montana. Moreover, we propose that the detritus from these organisms dominated the supply of contemporaneous biosiliceous sedimentation, fueling the production of spiculite deposits throughout the northeastern Panthalassic Ocean coast. We propose that the establishment and preservation of these *in situ* sponge meadows were controlled by bottom-water oxygenation and by hydrodynamic energy, respectively. We present evidence that sponges possibly demonstrated a partially infaunal life habit, leading to their misdiagnosis as trace fossils. These sponge body fossils frame the Phosphoria Rock Complex's transition from a starved, highly-concentrated phosphorite to a prolific glass ramp: an animal-mediated accumulation of opaline silica.

## Introduction

The Phosphoria Rock Complex (referred here simply as Phosphoria) [1] is a group of interrelated units of phosphorite, chert, and carbonate sedimentary rocks deposited in the western United States and is well-known as a vital phosphate resource [2–13] (Fig 1). In addition to its unique abundance of phosphorite, the Phosphoria also contains the southernmost extent of widespread epicontinental chert deposition during the mid-Permian. The Permian Chert Event [14] is a ~ 10–30 Myr interval dominated by biosiliceous sedimentation sourced from siliceous sponges and radiolarians, now preserved as chert strata in basins that rimmed the northwestern coast of Pangea: from

**Data availability statement:** All relevant data are within the paper and its Supporting Information files.

**Funding:** WM- Idaho State University Geosciences Geslin Award, https://www.isu.edu/geosciences/resources/endowments_grants_scholarships/Tobacco WM- Root Geological Society, www.trgs.org KR- ACS PRF 56988, American Chemical Society, https://www.acs.org/ ZW-Paleontological Society Student Research Grant.

**Competing interests:** The authors have declared that no competing interests exist.

the Phosphoria of the western USA, through the Sverdrup Basin of northern Canada, to the Kapp Starostin Formation of Svalbard, Norway (Fig 1A) [14–20].

The chert strata of these basins are not simply the result of waning phosphorite or lack of carbonate production, but are the sedimentary remains of a massive proliferation of organisms that produced siliceous biogenic sediments. Murchey [16] noted that biosiliceous sediment production was coeval with fertile overlying waters, conditions necessary to support phosphogensis [24]. Beauchamp and Baud [15] observed a wholesale transition from cool-water carbonate sediment factories that dominated the northwest coast of Pangea to biosiliceous sediment factories from the early to mid-Permian, respectively. Some researchers postulate this transition was caused by cold water temperatures [15,16] and possibly low pH [25], conditions which may have hampered carbonate-skeleton producing organisms in favor of biosiliceous. Others have suggested that the carbonate to biosiliceous faunal transition and the prolonged interval of biosiliceous dominance demonstrate hysteresis, a fundamental change in the ecological state of these marine embayments [19].

At times during the Permian Chert Event, the Phosphoria Sea favored the prolific growth of sponges, as evidenced by tens to over hundred meters-thick deposits of spiculitic chert [11,19,20,26,22]. Ritterbush [19] estimated that the Phosphoria glass ramp, a depositional system dominated by biosiliceous sediment production [17], may have buried as much as 2% of annual global silica inputs during the Permian via the accumulation of siliceous sponge spicules

The sheer volume of sponge spicule-rich rock retained in the Phosphoria glass ramp presents a puzzle. Where are all the fossil sponges of the glass factory? Two factors make the problem hard to resolve. First, demosponge body fossils have a low preservation potential: the spongin soft tissues rapidly decay and the interlocking spicule skeleton is easily disaggregated after death [27]; second, the opaline silica of which spicules are composed is subject to shallow burial diagenesis (dissolution, concentration, precipitation) and deep burial conversion to the more stable form of chert [28,29]. Because of this, preserving identifiable body fossils of demosponges requires unique physicochemical conditions.

Chert comprises nearly half the thickness of the Phosphoria in some areas [22], yet only small, isolated occurrences of intact sponge fossils are described [1,30,31]. By contrast, the contemporaneous Permian Basin in western Texas, south of the Phosphoria, preserves a great diversity of Permian sponges [32]. Herein, we argue that a curious lithofacies of cylindrical, vertically-oriented chert concretions preserves the cryptic remains of *in situ* sponge body fossils. Further, we reinterpret these structures as vast, level-bottom communities of tubular siliceous sponges that were pervasive, both spatially and temporally, throughout the extent of the Phosphoria. These enigmatic strata present a controversial piece to the regional puzzle. Previous studies interpreted these features as inorganic structures [33], trace fossils [34], and sponge body fossils [20]. We are motivated to reexamine these problematic features within the new depositional context of the glass ramp.

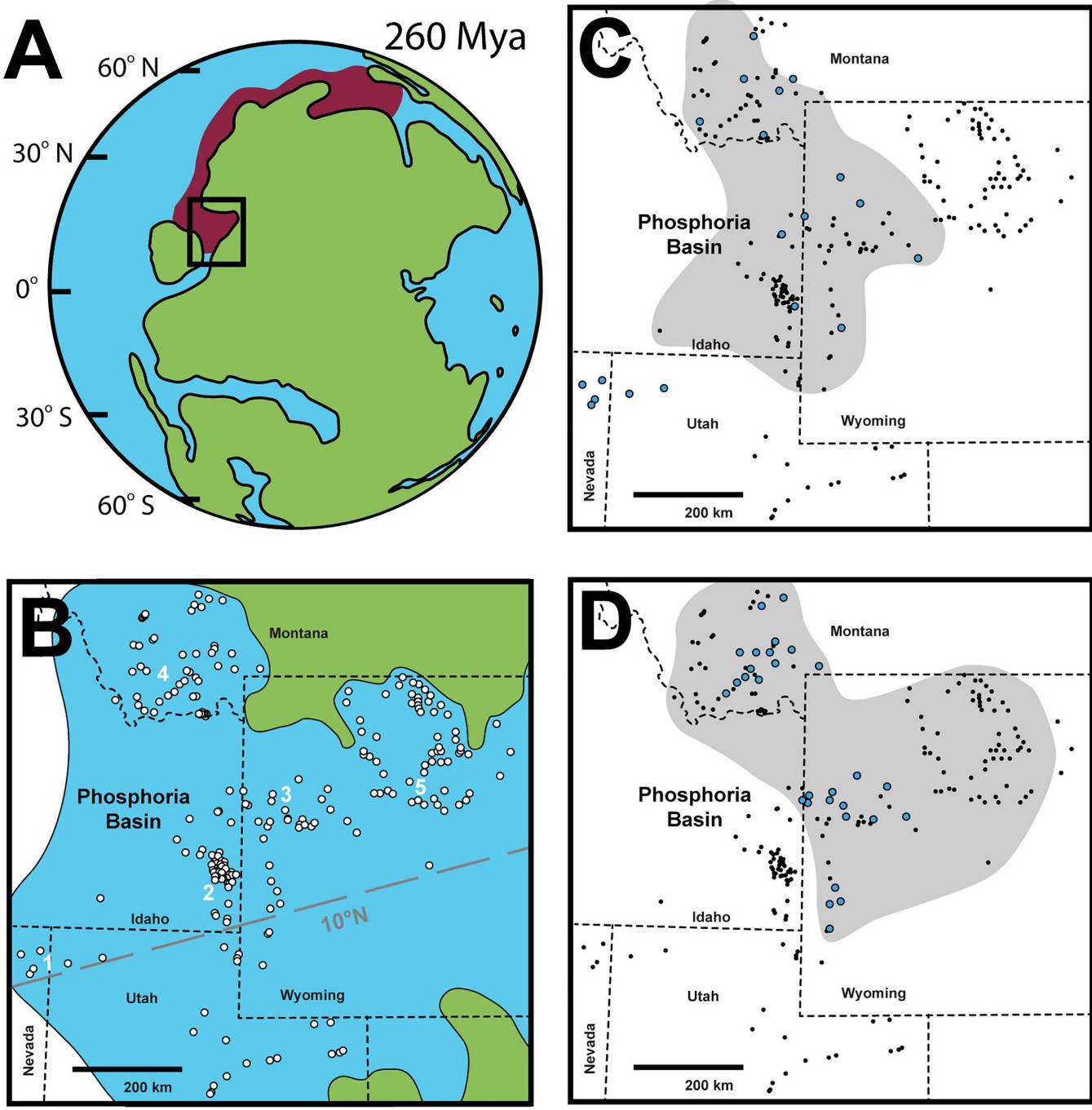

**Fig 1. Paleogeography of the Permian Chert belt and stratigraphic column locations visited in the field and from reports.** (A) Paleogeography of the Permian Chert belt (red) based on [15]; basemap of 260 mya PaleoDEM from GPlates [21]. (B) Locations of all stratigraphic columns referenced in collected reports of the Phosphoria Basin (white circles). This basemap is modified from [22] CEGA© [1994] and reprinted by permission of CEGA whose permission is required for further use. The green polygon is land, and the blue polygon is water; the white area is unknown as per [22]. See S1 Table for further locality information. White numbers correspond to locations of regionally representative stratigraphic columns (Fig 2). Figure originally modified from [10]; basemap from [23]; and, paleolatitude from [4]. (C) Locations of all stratigraphic columns coded by the presence (blue circle) or absence (black dot) of cylindrical chert concretions during the Franson depositional cycle. The gray polygon indicates the approximate extent of Rex Chert as assessed via stratigraphic columns from collected reports in this study. (D) Locations of all stratigraphic columns coded by the presence (blue circle) or absence (black dot) of cylindrical chert concretions during the Ervay depositional cycle. The gray polygon indicates the approximate extent of Tosi Chert as assessed via stratigraphic columns from collected reports in this study.

## Geologic setting

The Phosphoria Rock Complex encompasses all of the interrelated lithologic groups that comprise the Phosphoria Basin, from southern Montana to eastern Wyoming, to Idaho, to northern Utah, and to eastern Nevada (Fig 2) [3–5,7]. The Phosphoria is subdivided into three depositional cycles: the Grandeur, Franson, and Ervay. Each cycle is named for the carbonate member that caps the sequence. As data is limited for the Grandeur cycle, we focus our attention on the Franson and Ervay depositional sequences. The Franson and Ervay are retrogradational and are each composed of a transgressive phosphate-rich mudstone unit (i.e., Meade Peak and Retort, respectively), a spiculitic chert unit (i.e., Rex and Tosi, respectively), and a bioclastic carbonate (i.e., Franson and Ervay, respectively) or a sandstone unit (i.e., Shedhorn).

There is much debate as to the age of the Phosphoria [6,35,36], and a full discussion of the chronostratigraphy is beyond the scope of this paper, but the reader is referred to Matheson and Frank [11] for a more expansive treatment of these issues. Given the difficulty in age resolution, we have chosen to adopt the Wardlaw [36] age model, which is based upon conodont and brachiopod biostratigraphy from numerous stratigraphic sections from northeastern Nevada to western Wyoming, tracking across the depocenter of the Phosphoria. This age model places the Phosphoria deposition as the latest Kungurian to Wordian age.

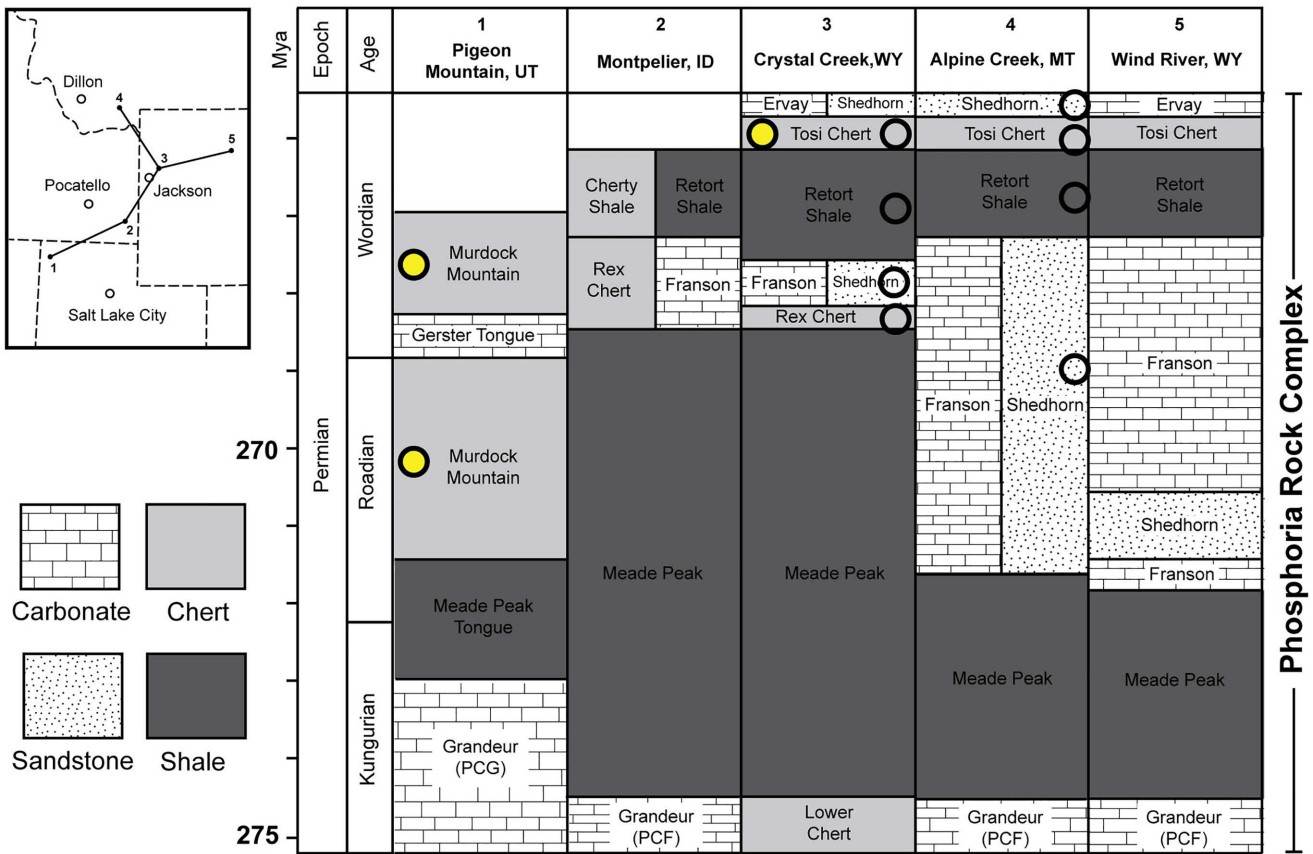

**Fig 2. Five representative stratigraphic columns of units across the Phosphoria Rock Complex.** Circle symbols indicate stratigraphic units in which cylindrical chert concretions are present in the region proximal to the column location. Open circles indicate reports of stratigraphic occurrence of chert cylinders; Yellow circles indicate units in which authors visited and sampled cylindrical chert concretions (See S1 Table for further locality information). This figure is modified from Cressman and Swanson [5] and includes stratigraphic data from [11,22].

Chert occurs in each formation of the Phosphoria Rock Complex, with notable concentrations in the southwestern extent of these strata. [1,3–6]. Although chert may be completely recrystallized, lacking any primary microfabrics, there is still ample petrographic evidence (i.e., presence of spicules) that siliceous sponges were primary sediment producers in both carbonate-rich and chert-rich strata [12,14,16,20]. Spiculitic cherts are most concentrated in the Rex Member (Fig 1C) and Tosi Member (Fig 1D) of the Phosphoria Formation, and the Murdock Mountain Formation of the Park City Group, where it is common to see sponge spicules in thin sections [11,12,20]. These strata have been interpreted as representing three glass ramp lithofacies: (1) Laminated black cherts, interpreted as offshore, low-energy, distal to sponge habitats (see facies S1b [20]; facies FA2-1 [12]); (2) massive, black to light cherts, interpreted as well-oxygenated subtidal sponge meadows (see facies S1a and S3 [20]; facies FA2-2 [12]); (3) irregular, nodular chert occurring in dolomitic, sandstone, and/or limestone host, interpreted as mixed deposition proximal to sponge meadows (see facies C1 [20]; facies FA2-3 [12]).

## Methods

We assess the outcrop expression and petrography of cylindrical chert concretions collected during field campaigns in eastern Nevada, northern Utah, and eastern Wyoming [20,37,38]. Wyoming concretion samples were collected by Moynihan [38] during summer 2017 from stratigraphic sections at Crystal Creek, Gros Ventre Slide, Flat Creek, and Ski Lake, WY. In total, 45 samples were collected, transversely and longitudinally slabbed, and 5 thin sections were cut. Nevada and Utah concretion samples were collected by [20,37] during field campaigns in the summers of 2016–2018, and 2020 from stratigraphic sections at Pigeon Mountain, UT, and Montello Canyon, NV. In total, 20 samples were collected, 3 were transversely and longitudinally slabbed, and 4 large-format (50 x 76 mm) thin sections were cut. Petrographic analyses included plane-polarized and cross-polarized light microscopy.

We used our lithologic and petrographic observations described above as a tool to search through extensive, historic lithologic descriptions from regional USGS reports of the Phosphoria region. We confidently identified stratigraphic sections and specific beds that contain similar reports of high-angle, cylindrical chert concretions (e.g., tubular cherts, p. 85–86 of [4]; columnar concretions, p. 351–354 of [5]). See S1 Table for locality, cylindrical chert concretion presence/absence, and stratigraphic unit data. See S2 Table for bed-specific lithologic descriptions of cylindrical chert concretions reported by [3–5]. See S3 File for additional references for data presented in S1 and S2 Tables.

## Results

### Cylindrical chert concretions

Vertical chert cylinders of the Phosphoria share similar morphology, dimensions, and internal microtextures across sites from our field observations and those of the USGS reports. They typically have an elongate conical form with a rounded base. The circular to oval cross-section (5–6 cm diameter) extends vertically for 10–150 cm (Fig 3A and 3B). The outer margin has a ringed or bulbous texture. Chert cylinders are densely packed with little (a few cm) to no matrix between discrete concretions (Fig 3A). The edges of cylinders abut one another, for their entire height, but these edges contain no flattened surfaces at their contacts, and no cylinder is observed cross-cutting another. The matrix surrounding chert cylinders is recessive weathering, and depending on the host rock, may comprise black chert, fine quartz sand, and mud, or minor amounts of carbonate grains that crumble when disturbed. The bases of cylinders are observed deflecting the underlying matrix downward ~5 cm below the cylinder.

Horizons of cylinders are frequently fractured, and bases of cylinders are generally broken off, forming a talus slope beneath the bedding horizon. When preserved, cylinders exhibit a conical-shaped base that protrudes in positive relief from the underside of the bedding horizon (Fig 3C). Bases of chert cylinders may be found at multiple levels throughout the bedding horizon. Variation in morphology includes stouter vase-shaped morphologies (20 cm diameter, 5–10 cm tall), and in some examples, "budding" is observed where one conical cylinder branches upward laterally from one below (Fig 3D).

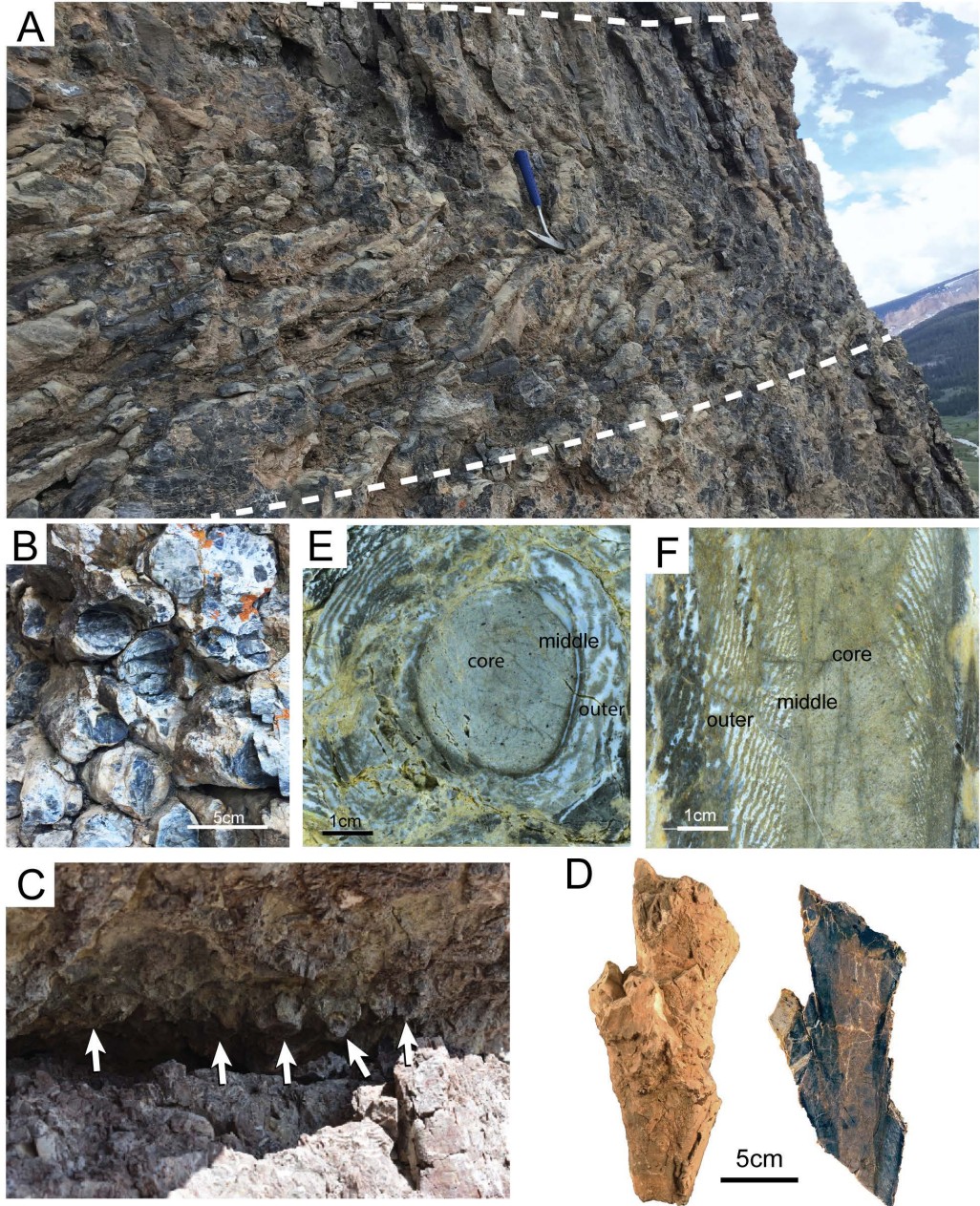

**Fig 3. Field and hand sample photos of vertical chert cylinders.** (A) Outcrop of densely packed sponges in bedded chert stratum, (B) with top-down, cross-sectional view; Tosi chert at Crystal Creek, WY. (C) Bases of sponges (white arrows) protruding from the underside of the bedding plane; Murdock Mountain Formation at Pigeon Mountain, UT. Nodules exhibit a conical shape base where they are not fractured. (D) UMNH.R.2016.67 sponge with budding, external view and longitudinal section; Murdock Mountain Formation, Montello Canyon, NV. (E) IMNH IP521−5 transverse slab cut, Flat Creek, WY. (F) IMNH IP522−14 longitudinal slab cut, Gros Ventre Slide, WY. Core, middle, and outer zones of the chert nodule are visible in slab cuts.

The cylinders are oriented, en-masse, vertical to bedding, and can be followed along strike for 100s of meters (~1 km at Crystal Creek, WY). Stratiform beds containing the cylinders range from 1 to 6 m in thickness. They are found in a variety of lithologies, including sandy dolostones and nodular carbonates, but they are most densely concentrated (up

to 20 cylinders per horizontal meter, ~100 per m²) in black, bedded cherts (Fig 3B). The total volume of a given horizon composed of chert concretions varies from 5% to 80% (See Table). In the Shedhorn Formation of southeastern Montana, Cressman and Swanson [5] found that some chert cylinders cross bedding planes of lithologically distinct strata, but this has not been observed elsewhere.

Thin section and slab-cuts of specimens, collected from western Wyoming and northern Utah, and Nevada, reveal that the vertical chert cylinders demonstrate a complex, but repeated, pattern of an inner "core" rich in sponge spicules, a "middle" with discontinuous bands of both chalcedony and microquartz, and cryptoquartz, and grading to an outer most zone of concentric bands of chalcedony, lacking spicules and preserving macroscopic zebra-textured chert (Figs 3E, 3F and 4).

The core of the cylinder is a 2–3 cm in diameter and composed of reddish to light-gray chert in slab-cut (Fig 3E and 3F). Thin section reveals the core is primarily composed of mixed micro- and crypto- quartz (Fig 4C and 4D). The core contains discrete particles, including opaque, subrounded phosphorite peloids (100–250 µm), micritized grains of possible biotic origin, and sponge spicules. (Fig 4C–4F). Sponge spicules are predominantly of monaxon morphology (~200 µm in length, 10–30 µm diameter), but triaxons spicules are also present (Fig 4F). In their study of lithofacies from the Mudrock Mountain Formation, Wistort et al. [20] observed triaxon spicules only in facies proximal to the source of spiculitic sediment (i.e., the sponge meadow biofacies). Spicules are commonly fractured, phosphatized, and may preserve the central axial filament. The spicules show no consistent internal lattice or arrangement, and appear of detrital origin. Likely, the sediment source was proximal to this facies as indicated by the preservation of more delicate triaxon spicules.

External to the core is a concentric "middle" zone that is less than 1 cm thick and darker in color than the core (Fig 4G–4L). The middle zone is easier to distinguish in thin section than in slab-cut hand samples. Thin section observation reveals that the macroscopic change in chert color corresponds to the separation of microquartz and cryptoquartz into discontinuous bands, and the addition of carbonate muds (Fig 4G and 4H). Spicules are present, but rarer than in the core zone. Light-colored bands of microquartz also contain spherulitic chalcedony, evidence of void-filling y (Fig 4I and 4J). These voids are interpreted as part of the primary texture because of the radial symmetry of the spherules and the length of the crystals (250 µm). Spherulitic chalcedony commonly fills voids in spiculitic cherts [28], radiating outward in all directions from a central nucleation point along the wall of the void. In the Montello canyon specimen, inclusions of dolomite are concentrated along the right edge of the middle layer (Fig 4K and 4L). Inclusions are euhedral to anhedral and exhibit sharp to irregular boundaries with the surrounding crystals (Fig 4). Some are closely associated with fractures filled with similar "dirty" dolomite [28]. These are likely to have occurred later in diagenesis than the initial precipitation of chalcedony.

The transition to the "outer" zone of the cylinder is marked by an absence of spicules, and by larger, repetitive bands (0.5 mm) of white chalcedony with brown chert and carbonate mud (Fig 4M and 4N). In slabbed specimens, the outer zone is easily recognizable as a zebra-texture chert which forms concentric, three-dimensional, upward-pointing conical texture (Fig 3F).

## Discussion

### Interpretation as remnants of sponge body fossils

Previous investigations over the last 70 years interpret vertical chert cylinders variously as inorganic structures, fossil burrows, or more commonly as of unknown origin [1,5,34,37,39]. Andersson and Sauvagnat [34] proposed that these cylinders were fossilized burrows and established a new ichnospecies, *Skolithos grandis*, suggesting that the concentration of spicules in the core of the tubular cherts was reworked particles, a common detrital component of Phosphoria cherts. They argue that faint, concave fabrics within the core are evidence for backfilling by a burrowing animal. Alternatively, we interpret these fabrics as spicules passively deposited in the spongocoel (central cavity) as the sponge grew upwards. The burrow hypothesis is untenable given that: (1) the high concentration of large dimension (100 cylinders/m², 3 cm diameter, >20 cm deep) open burrows would collapse in a poorly consolidated fine-grained matrix; and (2) at all

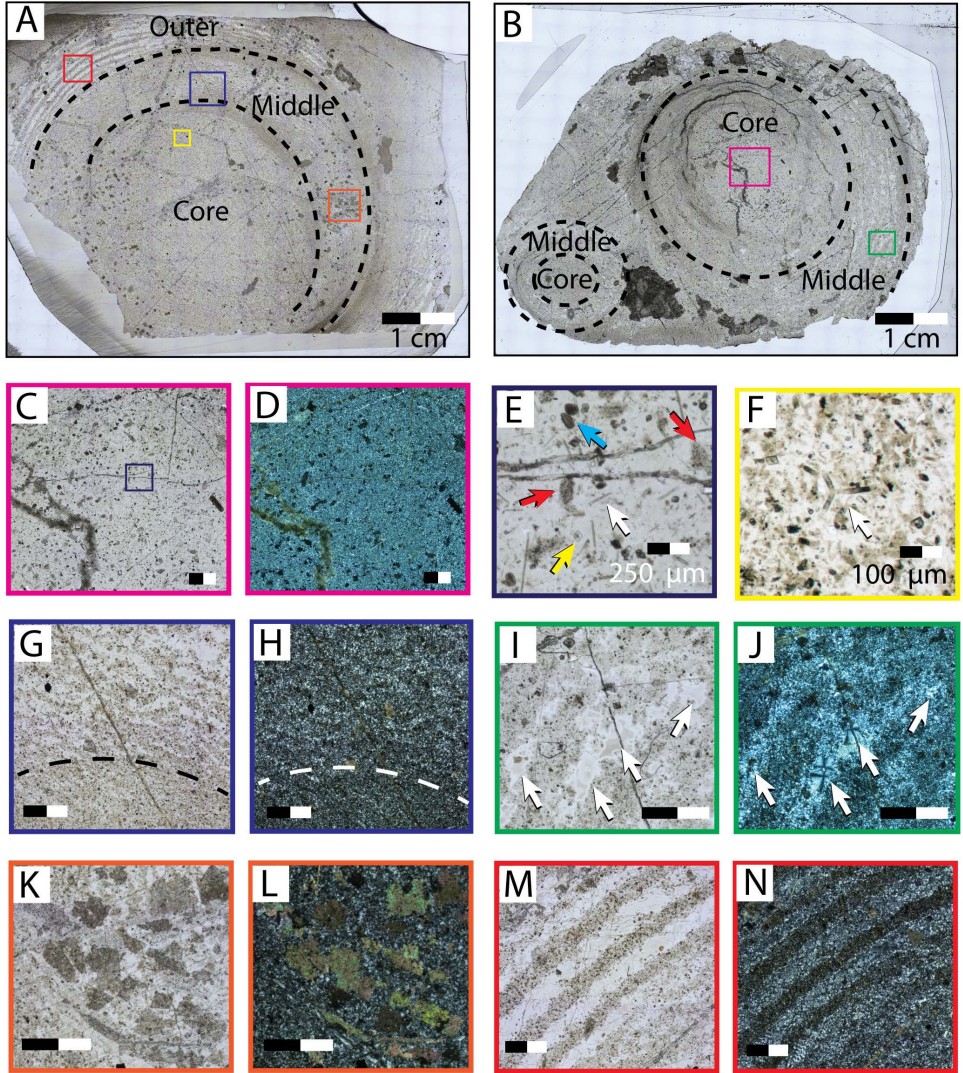

**Fig 4. Photomicrographs of the two vertical chert nodules cut transversely.** Large-format thin sections (50x76 mm) were made of specimens collected from the Murdock Mountain Formation at (A) Montello Canyon, NV, and (B) Pigeon Mountain, UT. The dashed lines indicate the transition between the core, middle, and outer textural zones. The Pigeon Mountain specimen has two distinct chert nodules, and no outer zone is present. The edge color of the boxes indicates the associated enlarged figure panel. All scales are 1 mm unless otherwise noted. (C & D) core zone, plane- and cross-polarized light, respectively. (E) enlarged core zone, plane-polarized light; Spicules (white arrow), saddle dolomite (yellow arrow), phosphatized peloids (blue arrow), and micritized carbonate grains (red arrows) are present; Many detrital monaxon spicules with random orientation are visible, but only one is indicated by the arrow for clarity. (F) a detrital triaxon spicule present in the core zone (white arrow). (G & H) middle zone, plane- and cross- polarized light, respectively; dashed lines indicate the border between the core and middle textural zones. (I & J) middle zone, plane- and cross- polarized light, respectively; Spherulitic chalcedony is present. (K & L) subhedral, "dirty" dolomite [28] in middle zone, plane- and cross- polarized light, respectively. (M & N) outer zone, plane- and cross- polarized light. Stark separation between microquartz and cryptoquartz forms a zebraic chert texture.

localities and units containing vertical chert cylinders, spicules are concentrated in the ~3 cm core, but not associated with surrounding or overlying matrix. The simplest solution is that the cylindrical concentration of detrital spicules owes to a discrete, proximal point-source, i.e., they are the remnants of *in situ* sponge bodies.

Our observations are best explained as *in situ* sponges that have retained their outer morphology, but whose spicules are partially disaggregated and diagenetically altered from silica mobilization during the early stages of diagenesis. This

interpretation is further supported by the microtextural zones present in the chert nodules. The "middle" zone, which contains rare spicules and evidence of void filling spherulitic chalcedony defines the outer edge of the sponge. We propose that fluids super-saturated in silica during early burial would have interacted with the decaying tissues of the sponge, providing a nucleation point for chalcedony to precipitate radially into the open voids of the sponge's internal canal network. The "outer" zone is a concretionary enlargement of the cylinder formed from the outward reprecipitation of diagenetic silica with zebraic textures [40]. In particular, morphologies demonstrating upward branching or budding whilst maintaining internal zonation of spicules argue strongly for a poriferan origin (Figs 3D and 4B). This zebraic texture likely occurred early in diagenesis as silica was mobilized from the skeleton of the sponge and possibly the surrounding sediments, diffusing outward and forming the concentric Liesegang-like banding pattern around the sponge's body [40]. Bands alternate between brown cryptoquartz, clear microquartz, and botryoidal chalcedony. The outer zone likely forms last of the 3 zones as the more segregated, larger microtextures of the clear microquartz indicate slower precipitation and conversion of opal-a to chert as compared to the middle and core

It is challenging to reconcile the tall, narrow dimensions of a chert cylinder as representing the free-standing height of an individual sponge above the sediment-water interface. We propose that the Phosphoria sponges had a psammbiotic life habit, growing < 30 cm above the sediment-water interface with the remainder of the sponge within the substrate. Psammbiosis is a habit utilized by some modern sponges to live within loose or unconsolidated sediments [41], often by agglutinating surrounding clasts to their ectosome [42]. There are modern observations of psammbiotic sponges forming sponge bioherms in the western Bahamas with living sponges as much as 2-meters below the sediment-water interface [43, p. 703–705]. There is precedent for psammbiosis among Paleozoic sponges, such as columnar aulacerid stromatoporoids [44]. Some aulacerids grew over 1 m in height and are typically 'decapitated' at the base, with the main portion of the column deposited parallel to bedding.

Preservation of sponges likely required a balance of sponge growth and sediment accumulation such that multiple generations of sponges could aggrade vertically, while the base was buried by sediments. The close packing of the sponges would have had a baffling effect, reducing the damaging effects of water energy and promoting the settling of sediments from the water column, facilitating *in situ* burial. Given the vertical continuity of the sponges, mixing by bioturbation was absent.

A possible paragenetic sequence of events is as follows (Fig 5): (1) Sponges colonize the seafloor; (2) Surrounding clasts are agglutinated into a sponge's outer most tissues helping to stabilize the animal's body; (3) Sponge growth maintains pace with sedimentation, interior of sponge may begin to fill with some disaggregated spicules and clasts from substrate; (4) The sponge animal is buried by surrounding sediments, filling the body chamber; (5) Opaline silica of the spicules mobilize and reprecipitate radially away from the core along a chemical gradient (e.g., Fig 4M and 4N); (6) Post-depositional tectonic deformation in Wyoming localities produces localized shear, shortening, and rotation of sponge cylindrical fossils (e.g., Fig 3A below hammer) [45].

**Extensive sponge meadows across the Phosphoria**

Phosphoria *in situ* sponges are preserved from Nevada northward to Montana from at least 47 localities and occur in the Murdock Mountain, Rex, Tosi, Lower Shedhorn, Upper Shedhorn, Ervay, and Retort members (Fig 2; S1 Table) [4,5,20]. These fossils are preserved in a variety of lithologies, including sandstone, sandy dolostone, nodular chert, carbonates, and bedded cherts. Black, bedded cherts of the Tosi, Rex, and Murdock Mountain members preserve the most densely packed and laterally extensive sponge meadow biofacies. The geographic pattern of preserved sponges generally follows an outboard arc that parallels the coastline of the Phosphoria Sea; however, this pattern may be obscured due to limitations of rock exposure and post-Paleozoic deformation of the stratigraphy (Fig 1). The arc of reported localities (Fig 1B) suggests a mid-ramp position of the preserved sponge meadow biofacies.

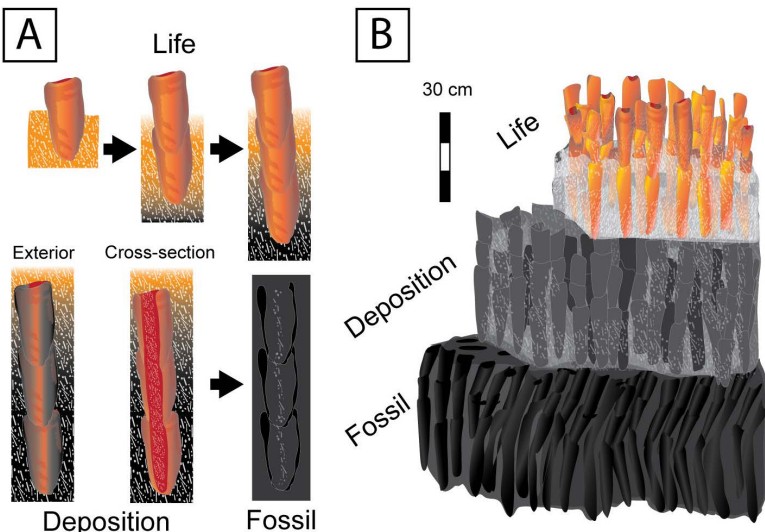

**Fig 5. Proposed model of sponge growth, burial, and fossilization within the sponge meadow biofacies.** (A) During life, the organism lives partially buried in the spiculitic sediments and grows vertically, keeping up with sediment input. During deposition, the sponge is buried either due to a high-energy event or the demise of the organism. The spongocoel is infilled with detrital particles of the surrounding environment. Tissues of the organism decay while pore space is filled by precipitation of aqueous silica sourced from the dissolution of the organism's skeleton and the surrounding sediments. (B) Paragenetic sequence diagram of sponge meadow diagenesis.

Interpolating from site observations, the sponge meadows are preserved over 630 km from north to south, representing a substantial biogenic system more than one quarter the size of the modern Great Barrier Reef. With a geographic span this size, and based on field observations of cylinder densities (~100 sponges/m²), sponges preserved in these meadows conservatively number in the billions. This unprecedented fossilized ecosystem represents ~15% of a much larger sponge glass factory whose remains are recorded by voluminous spiculitic cherts covering >220,000 km² of the Phosphoria Basin [19].

Sponge meadows are preserved on the ramp where their growth was in equilibrium with aggrading sediment. The prodigious release of spiculitic sediments and domination of level bottom environments by sponges would have created a peculiar marine ecosystem having no modern analogue. In dense concentrations, sponge meadows in the Phosphoria preserve few other benthos, either by calcifying epifauna or infaunal burrowers; however, onshore spiculites are commonly associated with bioturbation [12,17], and we observe brachiopods and foraminiferans within low-density sponge meadows (e.g., Rex Chert at Ski Lake). These observations support previous hypotheses that sponge recruitment and proliferation may produce negative feedback for carbonate producers [19] and that sponges and calcifying benthos are not mutually exclusive.

Stratigraphic sections with sponge concretions are present generally along a mid-ramp position between the basin depocenter in southeastern Idaho and the predominant shallow to supratidal deposits in central Wyoming and northeastern Utah (Fig 6) [4,5,12,20,37,38]. In these shallower strata, clean, winnowed, medium- to light- gray or purple spiculitic cherts and nodular cherts prevail [20]. We propose that the infrequent occurrence of sponge fossils in these shallowest strata is indicative of a hydrodynamic limitation to sponge preservation, a high-energy zone. This is likely the shallow boundary of sponge body fossil preservation. We postulate that the various types of bedded cherts present in the Phosphoria Rock Complex are various expressions of the sponge meadow biofacies, with their particular taphonomic expression controlled by hydrodynamic energy. Paleoenvironments above fairweather wave-base

produce the lighter-colored, nodular, sometimes chaotic cherts, and paleoenvironments at storm wave-base, or below, produce the laminated to massive, dark-gray to black cherts. In between these extremes, the *in situ* sponge meadow occurs.

## Implications for paleoceanography of the Phosphoria Basin

Much debate has surrounded the climatic and physicochemical patterns that fostered both the extensive deposits of biogenic chert and the concentration of economic phosphate in the Phosphoria Basin [7,46]. Phosphorite production is generally considered to require a high export of organic matter to the substrate via eutrophication; dysoxic conditions favorable for the precipitation of authigenic phosphorus minerals; and low sediment accumulation rates, concentrating phosphatic material [24].

In the Phosphoria Basin, an upwelling model is commonly invoked to explain the high primary productivity inferred to support such substantial phosphogenesis [3,4,7–9,12,13,47]. Modern examples of phosphogenesis generally require nutrient-rich, deep-water upwelling at the edge of the continental shelf [24]. However, the Phosphoria Basin exhibits a low-grade, ramp geometry and is likely shallow (<200 meters) [1], making modern shelf-edge phosphogenesis an unrealistic model [7,12,19]. To resolve ramp geometry with phosphogenesis, some researchers hypothesize the transport of an intermediate-depth, nutrient-rich water mass flowing from the north that would seasonally impinge on the Phosphoria Basin due to coastal upwelling via Ekman transport [8,12].

Periods of intermediate water mass transport would lead to eutrophication and increase organic matter production, as well as pervasive dysoxia and periods of anoxic and possibly euxinic conditions. During the deposition of the Meade Peak Member and Retort Member, both lowstand to transgressive intervals exhibiting peak phosphogenesis, body fossil collections tend to be low in diversity [1]. Trace fossils in these units are typical of the *Nerites* ichnofacies, generally correlated with dysoxic conditions (e.g., *Chondrites*, *Helminthopsis*) [13]. Lastly, geochemical proxies of organic carbon, total sulfur, and δ13C vary considerably across the shelf, indicative of variable dysoxic to periodically anoxic conditions [8,9]. Hiatt and Budd [8] hypothesize that oxygenation is the primary control on paleoecologic distribution during deposition of the Meade Peak Member.

Variation in oxygenation, similar to hydrodynamic energy described in the previous section, is a possible basinward control on the distribution, and subsequent preservation, of the *in situ* sponge meadow biofacies (Fig 6). Towards the depocenter of the basin (southeastern Idaho), periods of anoxia during lowstand and transgressive phases of peak phosphogenesis may have acted as a boundary to colonization by sponge communities. A singular report of sponge concretions from the Retort phosphatic shale in southwestern Wyoming is evidence of the inhibition of biotic recruitment. In southeastern Idaho, the only macrofauna present are low-diversity shell beds of *Orbiculoidea* and *Nucleopsis*, and fish bone lags [1]. This may represent a level of dysoxia in which sponges were unable to colonize. Although dysoxia probably continued to some extent, evidenced by continued phosphorite mineral production, bottom water oxygenation stabilized enough for broad sponge colonization during the transition to the Rex Member, Murdock Mountain Formation, and Tosi Member.

## Conclusions

Cylindrical chert concretions preserved in the Phosphoria, previously described as trace fossils, are reinterpreted as sponge body fossils. They preserve an extensive, 630 km midshelf section of the sponge meadow biofacies during both major depositional cycles of the Phosphoria Basin. This biofacies is one end-member of bio-siliceous deposition in the Permian Rock Complex and is distinguished from the more common bedded chert spiculites by means of lower overall hydrodynamic energy, allowing for their preservation. Due to sponge body dimensions, we speculate that these sponges had at least a partially psammbiotic life mode, an adaptation to living on

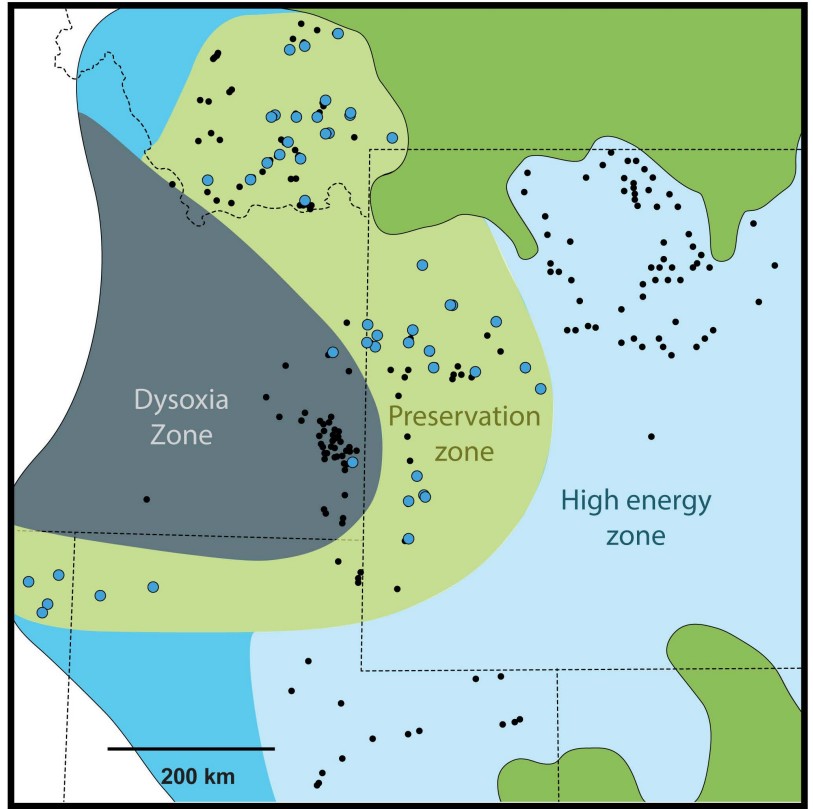

**Fig 6. Regional paleogeographic overview of the Phosphoria Basin modified from Fig 1.** We have overlaid all sponge occurrences from both the Franson and Ervay Cycles to which emphasizes the distinct zonation of sponge fossil occurrence. We speculate that in situ sponges are only preserved at a midramp preservational zone (green) due to high hydrodynamic energy in more onshore environments (light blue), and prevailing low oxygenation towards the basin depocenter (dark gray).

unconsolidated substrate. This biofacies was bounded by facies with extreme dysoxia and periodic anoxia, present in the deeper basin in southeastern Idaho, and by facies of higher hydrodynamic energy in central Wyoming.

## Supporting information

**S1 Table. This table contains the locality information for stratigraphic columns, stratigraphic unit information, and the presence of cylindrical chert concretions data used in this report.** Data from these columns was compiled from USGS reports [3–5,48–55], published articles [20,34], and a thesis [38]. Locality information recorded in the township and range format was converted to latitude and longitude using the GEOLocate web application (https://www.geo-locate. org/web/WebGeoref.aspx) [56].
(XLSX)

**S2 Table. This table contains the descriptions of beds in which vertical chert cylinders are present.** Descriptions were digitized and compiled from USGS reports [3–5,48–55].
(XLSX)

**S3 File. Additional references for data in S1 Table and S2 Table.**
(DOCX)

## Acknowledgments

We thank D. Wheeler for facilitating research in Wyoming (USDA Forest Service Paleo-0403-2017-02 permit). Wyoming specimens are reposited at the Idaho Museum of Natural History, Pocatello, USA. Utah and Nevada specimens are reposited at the Utah Museum of Natural History, Salt Lake City, USA. We thank Dr. Rossana Martini and an anonymous reviewer for their careful review of this manuscript.

*All necessary permits were obtained for the described study, which complied with all relevant regulations.*

## Author contributions

**Conceptualization:** Zackery Wistort, Leif Tapanila, William Moynihan, Kathleen Ritterbush.

**Data curation:** Zackery Wistort, William Moynihan.

**Formal analysis:** Zackery Wistort, Leif Tapanila, William Moynihan, Kathleen Ritterbush.

**Funding acquisition:** Zackery Wistort, William Moynihan, Kathleen Ritterbush.

**Investigation:** Zackery Wistort, Leif Tapanila, William Moynihan, Kathleen Ritterbush.

**Methodology:** Zackery Wistort, Leif Tapanila, William Moynihan, Kathleen Ritterbush.

**Supervision:** Leif Tapanila, Kathleen Ritterbush.

**Writing – original draft:** Zackery Wistort.

**Writing – review & editing:** Zackery Wistort, Leif Tapanila, Kathleen Ritterbush.

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
