## [Decision Letter · Decision Letter 0]

29 May 2025

PONE-D-25-21229Glass factory found: Basinwide (600 km) preservation of sponges on the Phosphoria glass ramp, Permian, U.S.A.PLOS ONE

Dear Dr. Wistort,

Thank you for submitting your manuscript to PLOS ONE. After careful consideration, we feel that it has merit but does not fully meet PLOS ONE’s publication criteria as it currently stands. Therefore, we invite you to submit a revised version of the manuscript that addresses the points raised during the review process.

We look forward to receiving your revised manuscript.

Kind regards,

Andrea Ceriani, PhD

Academic Editor

PLOS ONE

Journal Requirements:

WM- Idaho State University Geosciences Geslin Award, https://www.isu.edu/geosciences/resources/endowments_grants_scholarships/Tobacco

WM- Root Geological Society, www.trgs.org

KR- ACS PRF 56988, American Chemical Society, https://www.acs.org/

ZW-Paleontological Society Student Research Grant

3. We note that Figure 1 in your submission contain map images which may be copyrighted. All PLOS content is published under the Creative Commons Attribution License (CC BY 4.0), which means that the manuscript, images, and Supporting Information files will be freely available online, and any third party is permitted to access, download, copy, distribute, and use these materials in any way, even commercially, with proper attribution. For these reasons, we cannot publish previously copyrighted maps or satellite images created using proprietary data, such as Google software (Google Maps, Street View, and Earth). For more information, see our copyright guidelines: http://journals.plos.org/plosone/s/licenses-and-copyright.

Reviewers' comments:

Reviewer's Responses to Questions

**Comments to the Author**

1. Is the manuscript technically sound, and do the data support the conclusions?

Reviewer #1: Yes

Reviewer #2: Yes

2. Has the statistical analysis been performed appropriately and rigorously? 

Reviewer #1: N/A

Reviewer #2: N/A

3. Have the authors made all data underlying the findings in their manuscript fully available?

Reviewer #1: Yes

Reviewer #2: Yes

4. Is the manuscript presented in an intelligible fashion and written in standard English?

Reviewer #1: Yes

Reviewer #2: Yes

5. Review Comments to the Author

Reviewer #1: The manuscript by Wistort et al. submitted to PLOS One combined novel original lab/field work and interpretation with a detailed literature review. The authors propose a novel interpretation for a series of large-scale tubular features found throughout chert-bearing members of the Phosphoria Rock Complex. The authors interpretations are scientifically robust and sound and they are a new useful interpretation of these features in the emerging concept of glass ramps. As this is the core tenet of the manuscript, I commend the authors for this. As a result of this, I believe this paper is a valuable contribution to the literature on both the Phosphoria and glass ramps and look forward to seeing it published.

That being said, while the manuscript is generally well written, sufficiently organized and includes appropriate figures, there are a number of sections I believe the authors could ‘clean up’. Most of these involve relatively minor changes including adding additional detail, more appropriate use of citations, addressing of a few aspects of their model that aren’t addressed, fixing mistakes on figures, and clarifying sections that are not overly clear or have ambiguity. There are also a few aspects of the paper where I believe the authors need to revisit their data/figures in a more significant way, primarily regarding the lack of data in part of the landward Phosphoria Sea, as well as in the stratigraphy/age model used. Neither of these take away from or will modify the conclusions of the manuscript but will strengthen the robustness of the supporting data and background.

I believe that these suggestions will result in a ‘moderate’ amount of work to address these issues and look forward to seeing this work published in a stronger and more robust paper. I encourage the authors to view comments my comments as constructive criticism that is meant to help improve the manuscript that accompanies strong underlying scientific motivations and conclusions. I have provided extensive comments below.

Best,

Edward Matheson

Reviewer #2: Dear Authors,

The manuscript is well written, and the objectives are clearly presented. This is an important contribution, with strong regional relevance—although it is worth noting that the Phosphoria Rock Complex extends over a vast area (preserved from Nevada northward to Montana) and holds significant economic interest.

The general framework of the study is well established, and the methods applied to confidently reassign the cylindrical chert concretions preserved in the Phosphoria—previously described as trace fossils—as sponge body fossils are clearly explained and convincingly supported.

The results presented are compelling, and the model illustrating sponge meadows during life, after burial, and as fossilized remains is particularly helpful in supporting the interpretation of the vertical chert cylinders observed in outcrop.

Nonetheless, to better visualize the proposed, though speculative, life mode—at least a partially psammobiotic adaptation to life on unconsolidated substrates—additional clarification or illustration could further strengthen the argument.

I found the discussion on the possible depositional model for the Phosphoria Basin both interesting and convincing. The proposed interpretation—accommodating a low-gradient, ramp-like geometry and likely representing shallow water conditions (<200 meters)—is well supported and thoughtfully argued.

In conclusion, the figures are clear and sufficient, and the supplementary material provided is appropriate and helpful.

6. PLOS authors have the option to publish the peer review history of their article (what does this mean? ). If published, this will include your full peer review and any attached files.

**Do you want your identity to be public for this peer review?** For information about this choice, including consent withdrawal, please see our Privacy Policy .

Reviewer #1: No

Reviewer #2: **Yes: ** Rossana Martini

---

## [Author Response · Author response to Decision Letter 1]

22 Aug 2025

I have included a document which contains all responses to specific reviewer comments.

---

## [Editor Report · Decision Letter 1]

10 Sep 2025

Glass factory found: Basinwide (600 km) preservation of sponges on the Phosphoria glass ramp, Permian, U.S.A.

PONE-D-25-21229R1

Dear Dr. Wistort,

We’re pleased to inform you that your manuscript has been judged scientifically suitable for publication and will be formally accepted for publication once it meets all outstanding technical requirements.

Kind regards,

Andrea Ceriani, PhD

Academic Editor

PLOS ONE
---

## [Editor Report · Acceptance letter]

PONE-D-25-21229R1

PLOS ONE

Dear Dr. Wistort,

I'm pleased to inform you that your manuscript has been deemed suitable for publication in PLOS ONE. Congratulations! Your manuscript is now being handed over to our production team.

Kind regards,

on behalf of

Dr. Andrea Ceriani

Academic Editor

PLOS ONE